# Serum and Synovial Levels of Cathepsin G and Cathepsin K in Patients with Psoriatic Arthritis and Their Correlation with Disease Activity Indices

**DOI:** 10.3390/diagnostics13203250

**Published:** 2023-10-19

**Authors:** Stanislava Dimitrova Popova-Belova, Mariela Gencheva Geneva-Popova, Krasimir Iliev Kraev, Velichka Zaharieva Popova

**Affiliations:** 1Department of Propedeutic of Internal Diseases, Faculty of Medicine, Medical University of Plovdiv, Clinic of Rheumatology, University General Hospital “Sveti Georgi”, 4001 Plovdiv, Bulgaria; stanislava.popova@mu-plovdiv.bg (S.D.P.-B.); krasimir.kraev@mu-plovdiv.bg (K.I.K.); 2Department of Propedeutic of Internal Diseases, Faculty of Medicine, Medical University of Plovdiv, University General Hospital “Kaspela”, 4001 Plovdiv, Bulgaria; velichka.popova@mu-plovdiv.bg

**Keywords:** cathepsin G, cathepsin K, psoriatic arthritis

## Abstract

This retrospective case-control study examined the relationship between the serum and synovial levels of cathepsin G (CatG) and cathepsin K (CatK) in patients with psoriatic arthritis (PsA) and their association with disease activity. Methods: This case-control study involved 156 PsA patients, 50 patients with gonarthrosis (GoA), and 30 healthy controls. The target parameters were measured using enzyme-linked immunosorbent assay (ELISA) kits. The serum levels of CatG and CatK were found to be significantly higher in PsA patients compared to both control groups (*p* < 0.001). Moreover, they could distinguish PsA patients from healthy controls with 100% accuracy. Synovial fluid CatG and CatK were positively associated with the following indicators of disease activity: the VAS (rs = 0.362, rs = 0.391); the DAPSA (rs = 0.191, rs = 0.182); and the mCPDAI (rs = 0.378, rs = 0.313). Our results suggest that serum and synovial fluid CatG and CatK levels could serve as biomarkers for PsA. In PsA patients with synovial fluid crystals, elevated synovial CatG levels demonstrated a sensitivity of 89.54% and a specificity of 86.00% in distinguishing them from PsA patients without crystals. Similarly, elevated synovial CatK levels had a sensitivity of 93.67% and a specificity of 94.34% for distinguishing PsA patients with synovial fluid crystals from those without. Furthermore, the synovial fluid levels of both CatG and CatK showed positive associations with key indicators of disease activity, including the visual analog scale (VAS) (rs = 0.362, rs = 0.391), the disease activity in psoriatic arthritis (DAPSA) (rs = 0.191, rs = 0.182), and the modified composite psoriatic disease activity index (mCPDAI) (rs = 0.378, rs = 0.313). In conclusion, our findings suggest that the serum and synovial fluid levels of CatG and CatK hold promise as potential biomarkers for assessing disease activity in psoriatic arthritis.

## 1. Introduction

Cathepsins are lysosomal cysteine proteases that play crucial roles in the metabolism of both intracellular and extracellular proteins, both in healthy individuals and those with various diseases [1]. They are responsible for the degradation of the extracellular matrix and the processing of antigenic proteins [1,2].

In a study by Cheng et al. (2023), scientific evidence suggests that cathepsins are involved in the pathogenesis of vascular diseases, thrombosis, calcifications, and neovascularization [2]. Furthermore, the same authors concluded that cathepsins hold promise as valuable biomarkers in the patients they studied [2].

Psoriatic arthritis (PsA) is a chronic inflammatory joint disease that significantly affects the health-related quality of life of patients [1,3,4]. It is also associated with various comorbidities such as cardiovascular disease, metabolic syndrome, diabetes, and hyperuricemia [4,5]. Moreover, the investigation of different facets of the pathological processes in PsA has prompted researchers to explore biomarkers such as cytokines, chemokines, and cathepsins, which can be used to assess disease activity.

Cathepsin G (CatG) belongs to the serine protease family and was initially discovered in the azurophilic granules of neutrophil leukocytes, leading to its naming in 1976 [6]. CatG serves multiple functions, including its involvement in pathogen clearance; the regulation of inflammation through modifications of chemokines, cytokines, and cell surface receptors [7]; the activation of lymphocytes [8]; and the enhancement of epithelial cell permeability [9], among others.

CatG plays a pivotal role in the development and progression of certain autoimmune diseases. In patients with rheumatoid arthritis, both the concentration and activity of CatG are elevated in the synovial fluid when compared to healthy controls or individuals with osteoarthritis [10,11].

According to Velvart et al., articular cartilage from individuals with PsA did not exhibit staining when subjected to peroxidase–antiperoxidase and specific antibodies against CatG [11].

Cathepsin K (CatK), on the other hand, belongs to the papain-like cysteine protease family and serves as the primary specific protease found in osteoclasts and activated macrophages [12]. It is expressed in preosteoclasts, chondrocytes [13], endothelial cells, and more [13,14].

CatK plays a critical role in the degradation of the bone matrix. Furthermore, the regulation of osteoclastogenesis and the activity of cathepsin K, which contributes to various autoimmune and non-autoimmune diseases, are clearly influenced by various cytokines and hormones [15,16,17].

Figure 1 illustrates the role of cathepsins in inflammation-associated diseases and their involvement in cartilage damage (by Biasizzo M. et al.) [15].

The serum and synovial fluid levels of CatG and CatK have not been extensively studied in patients with PsA, despite their well-established roles in bone and cartilage damage in various diseases [18,19,20]. In the medical literature, there is a lack of sufficient research that demonstrates the role of synovial biomarkers in predicting more severe joint cartilage destruction in PsA compared to gonarthrosis (GoA). In our previous investigations, we found that 23.71% of PsA patients had the presence of crystals, primarily monosodium urate (MSU), in their synovial fluid, and none of the GoA patients had crystals (*p* < 0.001) [6]. The presence of MHY crystals was associated with a more severe clinical course of the disease and greater synovial involvement in the affected joints [6].

Another underexplored aspect is how the presence of MHY crystals in the synovial fluid of PsA patients influences the levels of CatG and CatK. Furthermore, the relationship between disease activity, assessed through disease activity indices, and the levels of CatG and CatK in the serum and synovial fluid of PsA patients require further investigation.

The primary objectives of this study were twofold: (1) to evaluate and compare the serum and synovial levels of CatG and CatK in patients with PsA, a control group of GoA patients, and healthy control subjects (only serum samples were tested in the latter group) and (2) to examine the correlation between the serum and synovial levels of CatG and CatK in PsA patients and various disease activity indices, including the visual analog scale (VAS), the disease activity in psoriatic arthritis (DAPSA), the psoriatic arthritis disease activity score (PASDAS), the modified composite psoriatic disease activity index (mCPDAI), and the health assessment questionnaire disability index (HAQ-DI).

This study aims to provide valuable insights into the potential role of CatG and CatK as biomarkers for disease activity in psoriatic arthritis.

## 2. Materials and Methods

### 2.1. Patients and Study Groups

This study was conducted as a retrospective case-control study. All participants had been diagnosed with PsA and GoA and received treatment at the Departments of Rheumatology at the University General Hospital “St. George” and the University General Hospital “Kaspela” in Plovdiv, Bulgaria. Data collection occurred between July 2020 and August 2022. Only PsA patients with peripheral polyarticular asymmetrical joint involvement were included, as they constituted the majority of individuals seeking treatment at the hospital during the recruitment period. Due to the limited number of patients with other PsA subtypes, it was not feasible to create representative subgroups.

The inclusion criteria for PsA patients were as follows: (1) a confirmed diagnosis of psoriatic arthritis with synovial effusion; (2) PsA patients not treated with a biological agent, such as TNF-α-blockers, IL-17 blockers, or IL-12/23 blockers; (3) the absence of mental health comorbidities; and (4) signed informed consent for participation in the study.

The exclusion criteria for the PsA patients were as follows: (1) patients under 18 years of age; (2) a diagnosis of a rheumatic disease other than PsA; (3) decompensated cardiovascular, pulmonary, or renal failure; and (4) pregnant or lactating women.

The GoA patients were included if they met the following criteria (ACR, 1991): (1) knee pain for at least 5 years, an age over 50 years, stiffness in the knee joint lasting less than 30 min, the presence of crepitations in the knee, a joint deformity, and enlargement without associated joint warmth; (2) radiographic evidence of osteophytosis of the knee joint; (3) an erythrocyte sedimentation rate <40 mm/h and a negative rheumatoid factor; and (4) a signed informed consent form for participation in the study. Excluded from the study were patients who met any of the following criteria: (1) the presence of crystalline arthropathy; (2) the presence of decompensated cardiovascular, pulmonary, renal, or hematological diseases; or (3) the presence of immunological phenomena.

The healthy control group consisted of individuals who exhibited no complaints or physical findings related to their internal organs, skin, or joints. They also had normal paraclinical parameters, were in good mental health, and provided written consent to participate in the study. The sera of the healthy control subjects were utilized exclusively for the assessment of serum CatG and CatK.

The study was conducted in strict accordance with the World Medical Association Declaration of Helsinki (1964) and its revised version (Edinburgh, 2000). All aspects of the study, including the analysis of patient data, the blood collection, the synovial fluid aspiration, and the content of the informed consent form, received approval from the Committee for Scientific Ethics at the Medical University of Plovdiv under protocol No. 4/10.06.2021.

### 2.2. Procedure for Evaluation of CatG and CathK in Serum and Synovial Fluid

Venous blood samples were collected in heparin sodium tubes, followed by centrifugation for 15 min at 1000× *g* to obtain plasma samples, which were then stored at −85 °C until the assays were performed. CatG was quantified using the Human CatG ELISA Kit, manufactured by Wuhan Fine Biotech Co., Ltd., Wuhan, China, with Cat. No. EH1903. The intra-assay coefficient of variation (CV%) for CatG is reported to be 5.1% within a mean concentration range of 0.156–10 ng/mL, with a sensitivity of 0.10 ng/mL.

The CatK levels were quantified using the Human CatK ELISA Kit (Cat. No. EH1908), also manufactured by Wuhan Fine Biotech Co., Ltd., Wuhan, China. The intra-assay coefficient of variation (CV%) for CatK was reported to be 4.3%, and this measurement was conducted within a mean concentration range of 0.095–10 ng/mL, with a sensitivity of 0.10 ng/mL.

The assay utilized two primary antigen-specific antibodies. Initially, the specific antibody was loaded into a microplate, which was composed of twelve separate strips, each equipped with eight wells. After determining the required number of standard and control samples, the relevant strips were extracted from their aluminum packaging, while the remainder were stored at −20 °C for preservation.

Following the manufacturer’s instructions, the lyophilized standard was appropriately diluted, and both the required standard and negative control samples were prepared. Subsequently, the second specific antibody (biotin-conjugated) and the peroxidase-conjugated secondary antibody were diluted to a ratio of 1:100 using their respective diluents.

After measuring the optical density, the duplicate values obtained from the standard and tested samples were averaged. We utilized the Curve Expert 4 software, ver. macOS 10.13 to create the standard curve. Once the standard curve was established, we calculated the concentration of each sample and applied the relevant dilution factor. Duplicate samples were assessed for each patient [21].

### 2.3. Arthrocentesis and Clinical Evaluations

In accordance with antiseptic guidelines, arthrocentesis on the patients with PsA presenting with hydrops in the knee joint was conducted by a rheumatologist. The synovial fluid obtained through arthrocentesis was subsequently examined independently by two rheumatologists using a Leica polarization microscope.

The patients’ disease state was evaluated using several assessment tools: the visual analog scale for pain intensity (VAS), the disease activity for psoriatic arthritis (DAPSA), the psoriatic arthritis disease activity index (PASDAI), the composite psoriatic disease activity index (mCPDAI), and the health assessment questionnaire disability index (HAQ-DI) [22,23,24,25].

For the DAPSA, PASDAI, mCPDAI, and HAQ-DI, the assessments were performed by two independent rheumatologists.

Based on the VAS pain score, the patients were categorized into three groups: patients with moderate pain (40–60 mm), patients with severe pain (60–80 mm), and patients with very severe pain (>80–100 mm) [22]. 

According to the DAPSA, the categorization of disease activity was as follows: low, ≤14; moderate, >14 to ≤28; and high, >28 [23].

For the PASDAI, the disease activity categories were: remission, <1.9; low disease activity, >1.9 to 3.2; moderate disease activity, >3.2 to 5.4; and high disease activity, >5.4 [24].

Based on the mCPDAI, the disease activity was categorized as: low disease activity, 1 to 3; moderate disease activity, >3 to 9; or high disease activity, >9 [25].

The HAQ-DI disease activity ranges included: 0 to 1, indicating a mild to moderate disability; 1 to 2, indicating a moderate to severe disability; and 2 to 3, indicating a severe to very severe disability [21].

All the tests were conducted on the same day, ensuring consistency and accuracy in the assessment process.

### 2.4. Statistical Analysis

The statistical analyses were conducted using SPSS version 26.0 (SPSS Inc., Chicago, IL, USA). The following methods were employed for different types of variables: Continuously measured variables were assessed for normality using the Shapiro–Wilk test. Normally distributed data were summarized with means and standard deviations (SDs), and comparisons were made using an independent-sample *t*-test. Non-normally distributed variables were presented as medians and interquartile ranges (IQRs) and analyzed using the Mann–Whitney U test. Categorical and ordinal data were presented as frequencies and percentages. Associations among categorical data were examined using the chi-squared test. Additionally, an ROC (receiver operating characteristic) analysis was employed to assess the results. Spearman’s rank-order correlations were calculated to examine the relationships between variables. All the statistical tests were two-tailed, and significance was determined at a type-I-error alpha level of 0.05 (*p* < 0.05).

## 3. Results

### 3.1. Demographic and Clinical Data for the PsA and GoA Patients

This study included a total of 156 patients diagnosed with PsA based on the CASPAR criteria, along with 50 patients diagnosed with GoA.

Among the PsA patients, a significant majority were men (*n* = 93, 59.60%), with an average age of 58.45 ± 11.23. In contrast, women constituted a smaller proportion (*n* = 63, 40.40%) with an average age of 53.45 ± 10.56. Additionally, the PsA patients had a notably higher body mass index (BMI) and a higher prevalence of various comorbidities, including diabetes, ischemic heart disease, hyperuricemia, dyslipidemia, and obesity, when compared to the GoA patients (Table 1).

Crystals were identified in the synovial fluid of 37 PsA patients, whereas none of the GoA patients exhibited crystals in their synovial fluid. The presence of crystals in the synovial fluid was significantly associated with PsA in contrast to the GoA group, where no crystals were detected (*p* < 0.001) (Table 1).

### 3.2. Assessment of PsA Patients versus a Control Group of GoA Patients and the Healthy Controls

The serum CatG levels in both the PsA patients and the GoA patients did not exhibit a normal distribution within all groups (*p* ≤ 0.001 for all six Shapiro–Wilk tests). Before comparing the groups based on the target parameters using the Kruskal–Wallis test, we conducted a multiple linear regression analysis to assess the influence of age and BMI as covariates, with the group as an independent variable. The analysis revealed that the effect of these covariates was not significant for CatG (age, *p* = 0.141; BMI, *p* = 0.891). Thus, age and BMI were determined not to have an impact on the CatG levels.

A subsequent analysis demonstrated a significant increase in the serum CatG levels among patients with PsA compared to the control groups (*p* < 0.01 for both GoA patients and healthy controls). Specifically, the results were as follows: the PsA patients had a level of 1.08 ± 0.04 (x ± Se), with a median of 1.09 ng/mL and an IQR of 14.04; the GoA patients had a median of 0.43 ng/mL with an IQR of 7.23; and the healthy controls had a median of 0.02 ng/mL with an IQR of 3.25 (Figure 2a).

The serum CatK levels in both the PsA patients and the GoA patients did not exhibit a normal distribution within all groups (*p* ≤ 0.001 for all six Shapiro–Wilk tests). To assess the influence of potential covariates, namely age and BMI, a multiple linear regression analysis was conducted with the group as an independent variable. The results of this analysis indicated that the covariates had no significant effect on the CatK levels (age, *p* = 0.157; BMI, *p* = 0.822).

The subsequent analysis revealed a significant elevation in the serum CatK levels among patients with PsA when compared to the control groups. Specifically, the results were as follows: the PsA patients had a level of 0.874 ± 0.08 (x ± Se), with a median of 0.589 ng/mL and an IQR of 23.04; the GoA patients had a median of 0.21 ng/mL with an IQR of 9.44; and the healthy controls had a median of 0.02 ng/mL with an IQR of 2.96 (Figure 2b).

Based on the levels of CatG in the serum, psoriatic arthritis (PsA) patients could be distinguished from healthy controls with 100% accuracy when the criterion value for CatG was >6 ng/mL. When distinguishing PsA patients from gonarthrosis (GoA) patients, the diagnostic accuracy was 61.76% when the criterion value for CatG was >1.01 ng/mL.

The serum CatK levels allowed for the differentiation of PsA from GoA patients with 100% accuracy at a criterion value of CatK > 0.86 ng/mL. A slightly lower accuracy rate (62.94%) was observed when distinguishing PsA patients from healthy controls at a criterion value of CatK > 0.73 ng/mL (Table 2).

#### The Levels of Synovial Fluid CatG and CatK in Patients with PsA and GoA

The PsA patients were divided into two groups based on the presence or absence of MSU crystals in the synovial fluid. In the group of PsA patients, those with crystals exhibited higher synovial CatG levels (median = 3.23 ng/mL; IQR = 23.01) compared to PsA patients without crystals (median = 2.41 ng/mL; IQR = 12.76, *p* = 0.04) and GoA patients (median = 0.34 ng/mL; IQR = 0.51, *p* < 0.02). Furthermore, the synovial CatG levels in PsA patients without crystals were significantly elevated compared to those in GoA patients (*p* < 0.001) (Figure 2a).

Similarly, PsA patients with crystals in their synovial fluid had significantly higher synovial fluid CatK levels (median = 2.32 ng/mL; IQR = 2.69) compared to PsA patients without crystals (median = 0.09 ng/mL; IQR = 0.42, *p* = 0.001) and GoA patients (median = 0.04 ng/mL; IQR = 0.21, *p* = 0.01). Notably, the median CatK level in PsA patients without crystals was identical to that in GoA patients (*p* = 1.000) (Figure 2b).

The synovial fluid CatG levels demonstrated 100% accuracy in distinguishing PsA patients, both with and without crystals, from GoA patients, using a criterion value of CatG > 2 ng/mL. The ability of synovial fluid CatG to discriminate between PsA patients with crystals and those without crystals was 91.08% (Table 3).

Synovial fluid CatK showed different diagnostic accuracies in PsA patients with and without crystals. PsA patients with crystals were discriminated from GoA patients with an accuracy of 64.44% at a criterion value of CatK > 0.51 ng/mL.

Notably, synovial fluid CathK had the same accuracy in discriminating PsA patients with crystals from both GoA patients and PsA patients without crystals, achieving an accuracy of 71.23% at the same criterion value of CatK > 0.51 ng/mL (Table 3).

### 3.3. The Relation of Serum and Synovial CatG and CatK Levels in Patients with PsA to Disease Activity

The VAS categories demonstrated a significant positive association with the levels of CatG (rs = 0.362, *p* < 0.001) and CatK (rs = 0.391, *p* < 0.001) in the synovial fluid (Table 4).

According to the DAPSA index, 71 (45.50%) of the PsA patients exhibited low disease activity, 38 (24.40%) were associated with moderate disease activity, and 47 (30.10%) were diagnosed with high disease activity. The DAPSA categories exhibited a significant positive relationship with both the serum and synovial fluid levels of CatG and CatK (*p* < 0.001) (Table 4).

The PASDAI categories displayed a significant positive relationship solely with the synovial fluid levels of CatK (rs = 0.384, *p* < 0.001).

The mCPDAI categories exhibited a significant positive relationship with the synovial fluid levels of CatK (rs = 0.378, *p* < 0.001), but not with the serum and synovial fluid levels of CatG (*p* > 0.05).

The HAQ-DI categories displayed a significant positive relationship with the serum levels of CatG (*p* < 0.05) and the synovial fluid levels of CatK (*p* < 0.001).

## 4. Discussion 

The serum and synovial fluid levels of CatG and CatK have not been extensively studied in patients with PsA, despite their well-established roles in bone and cartilage damage in various diseases [9,16,19,26].

While the specific role of these cathepsins requires further elucidation, it may involve the activation of immune cells and the mobilization of the immune response by inducing the production of lymphokines. These, in turn, promote T-cell-dependent cellular immunity and the production of antigen-specific antibodies [9,16,19,26].

In some rheumatological diseases, such as rheumatoid arthritis (RA), the concentration and activity of certain cathepsins, such as CatC, are increased in synovial cells, synovial fluid, and even in sera. The inhibition of these cathepsins using specific inhibitors has shown promise in suppressing autoimmune joint inflammation and osteoclastic bone resorption, leading to reduced cartilage damage [27,28]. According to Skoumal et al., RA patients with elevated levels of CatK tend to experience more severe joint destruction [28].

This is our first report on the serum and synovial fluid levels of CatG and CatK in patients with PsA. We investigated these cathepsins as part of a study of different serum and synovial markers of activity in PsA patients because they are associated with bone and extracellular matrix degradation as well as inflammatory processes.

According to Veale et al., improving our understanding of the pathogenesis of PsA may help establish validated biomarkers for diagnoses, predictions of the therapeutic response, and remission [29].

We did not find a study that examined the serum and synovial fluid CatG and CatK levels in patients with PsA and compared them with bony gonarthrosis and healthy controls.

In our study, the serum and synovial fluid CatG and CatK levels demonstrated an accuracy of 100% in differentiating PsA patients from healthy controls, regardless of age and BMI. Another challenge for clinicians is distinguishing PsA from other joint diseases such as GoA.

According to Huang et al., CatG may be involved in the degradation of proteins in the synovial fluid of patients. The authors detected endogenous CatG in 16 patients with knee OA without describing its exact concentration and proved that its high amount affected pathological processes in degenerative joint disease [30].

Both serum CatG and CatK demonstrated diagnostic potential in our study in distinguishing PsA from GoA patients with a sensitivity of 100%. The PsA patients had higher serum CatG and CatK than the GoA patients.

On the other hand, serum CatG and CatK did not show strong associations with disease activity in the patients with PsA, as assessed through the VAS, PASDAI, and mCPDAI. 

In addition, serum biomarkers such as CatG and CatK can help with the correct diagnosis and treatment of PsA.

We obtained encouraging results for assessing the levels of cathepsins G and K in the synovial fluid of patients with PsA. We proved that both levels are higher than those of patients with GoA. Our results were even more revealing when we divided the patients with PsA into two groups: those with MNU crystals and those without crystals.

The PsA patients with crystals were distinguished from the GoA patients with an accuracy of 100%, whereas cathepsins G and K were not found to be reliable diagnostic biomarkers of PsA patients versus GoA patients. Both the PsA and GoA patients in our study had active disease, and the presence of crystals in some of the PsA patients was the only other factor. The low synovial cathepsin G and K levels in the PsA and GoA patients with no crystals suggest that elevated cathepsin G and K levels may be attributed to crystals rather than disease progression.

Regarding the relationship with disease activity indices, synovial CatG and CatK revealed positive correlations with the VAS, DAPSA, mCPDAI, and HAQ-DI. This study’s findings suggest that elevated levels of these cathepsins are associated with higher disease activity, as indicated by the disease activity indices. This information underscores the potential clinical relevance of monitoring CatG and CatK levels as biomarkers for disease activity in PsA. The early detection of higher cathepsin levels may indicate the need for more aggressive and timely treatment. This could be valuable information for rheumatologists and healthcare providers involved in managing patients with PsA, as it may help guide treatment decisions and improve patient outcomes.

### Limitations of the Study

The results of this investigation were limited to patients over the age of 18. Moreover, only arthrocentesis of the knee joint was examined.

## 5. Conclusions

An assessment of the serum and synovial levels of cathepsin G and cathepsin K in patients with PsA versus a control group of GoA patients proved that the levels of cathepsins are higher in the serum and synovial fluid of patients with PsA. This is associated with more severe destruction in PsA patients and, as such, these cathepsins may be used as diagnostic biomarkers.

The relation of the serum and synovial levels of cathepsin G and cathepsin K in patients with PsA to disease activity indices, including the VAS for pain intensity, the DAPSA, the PASDAS, the mCPDAI, and the HAQ-DI, showed that higher levels of the studied cathepsins were associated with higher disease activity, and as such, they may be used as prognostic biomarkers.

## Figures and Tables

**Figure 1 diagnostics-13-03250-f001:**
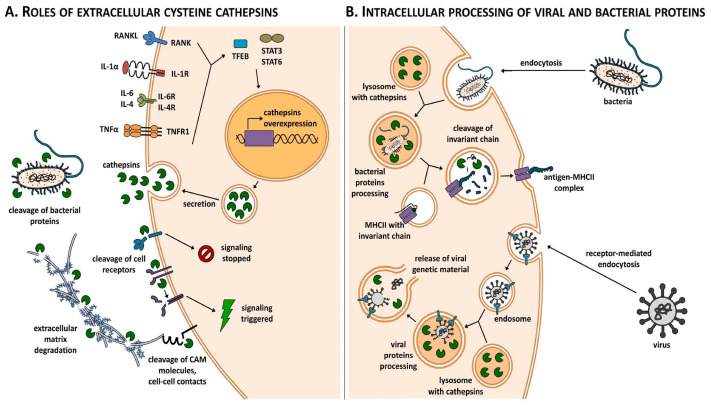
Cathepsins in inflammation-associated diseases and in cartilage damage (by Biasizzo M. et al., in “Cysteine cathepsins: A long and winding road towards clinics”, Mol. Aspects of Medicine, Vol. 88, 2022,101150,ISSN 0098-2997) [15].

**Figure 2 diagnostics-13-03250-f002:**
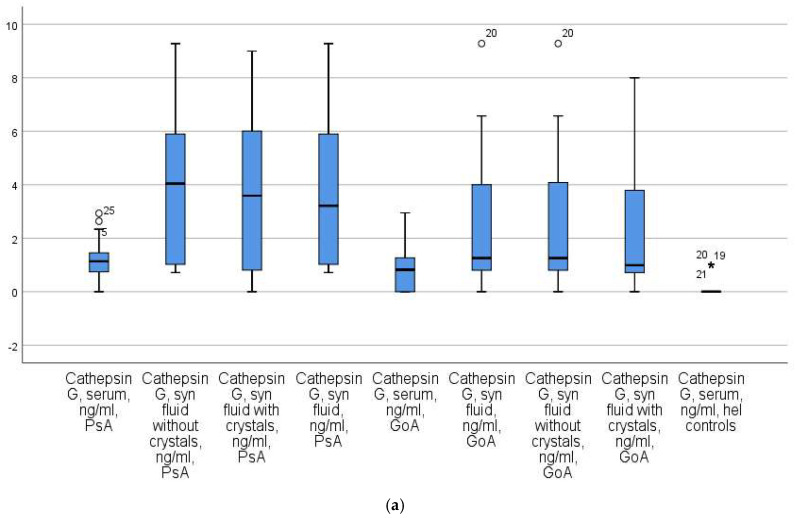
(**a**) Boxplots of serum and synovial fluid cathepsin G across the study groups--patients with PsA, patients with GoA, and the healthy controls. (**b**) Boxplots of serum and synovial fluid cathepsin K across the study groups- patients with PsA, patients with GoA, and the healthy controls. *—unit value.

**Table 1 diagnostics-13-03250-t001:** Demographic and clinical data for the PsA and GoA patients.

	Groups				
Variables	PsA Patients (*n* = 156)		GoA Patients (*n* = 50)		*p*
Sex *n* % ◯ Men *n* (%) ◯ Women *n* (%)	93	59.60	25	50.00	
	63	40.40	25	50.00	0.253 ^f^
Age (years), mean ± SD	54.36 ± 13.55		62.46 ± 11.55		<0.001 ^t^
Men	58.45 ± 11.59		66.01 ± 9.16		
Women	53.78 ± 10.66		59 ± 9.34		
BMI, mean ± SD	33.40 ± 5.44		28.36 ± 4.33		<0.001 ^t^
Comorbidity *n* %					
Hypertension	112	71.70	41	82.00	0.193 ^f^
Diabetes	125	80.12	13	26.00	<0.001 ^f^
Ischemic heard diseases.	61	39.10	9	18.00	0.006 ^f^
Hyperuricemia	111	71.15	3	6.00	<0.001 ^f^
Dyslipidemia	99	63.46	13	26.00	<0.001 ^f^
Obesity	132	84.60	25	50.00	<0.001 ^f^
Synovial fluid crystals, *n* %	37	23.70	-	-	<0.001 ^f^

^t^—independent sample *t*-test; ^f^—Fisher’s exact test.

**Table 2 diagnostics-13-03250-t002:** Results from the ROC curve analysis regarding the ability of serum cathepsin G and cathepsin K to discriminate PsA patients from healthy controls and GoA patients.

Variables	AUC 95% CI	*p*	Criterion Value	Sensitivity	Specificity
Cathepsin G, serum					
PsA vs. healthy controls	1.00 (1.00 to 1.00)	<0.001	>6	100%	100%
PsA vs. GoA	0.897 (0.812 to 0.946)	<0.001	>61.76	91.08%	64.00%
Cathepsin K, serum					
PsA vs. healthy controls	1.00 (1.00 to 1.00)	<0.001	>5	100%	100%
PsA vs. GoA	0.897 (0.798 to 0.841)	<0.001	>62.94	90.71%	64.00%

ROC curve—receiver operating characteristic curve; AUC—area under the curve; PsA—psoriatic arthritis; GoA—gonarthrosis.

**Table 3 diagnostics-13-03250-t003:** Results from the ROC curve analysis regarding the ability of synovial fluid cathepsin G and cathepsin K to discriminate patients with PsA, with and without crystals, from GoA patients.

Variables	AUC 95% CI	*p*	Criterion Value	Sensitivity	Specificity
Cathepsin G in synovial fluid					
PsA with crystals vs. GoA	1.00				
	(1.00 to 1.00)	<0.001	>2	100%	100%
PsA without crystals vs. GoA					
	1.00				
	(1.00 to 1.00)	<0.001	>2	91.08%	74.00%
PsA with crystals vs. PsA without crystals					
	0.657				
	(0.612 to 0.754)	<0.001	>14	89.54%	86%
Cathepsin K in synovial fluid					
PsA with crystals vs. GoA					
	1.00				
	(1.00 to 1.00)	<0.001	>5	100%	100%
PsA without crystals vs. GoA					
	1.00				
	(1.00 to 1.00)	<0.001	>65.44	90.71%	64.00%
PsA with crystals vs. PsA without crystals					
	0.897				
	(0.687 to 0.891)	<0.001	>71.23	93.67	94.34

ROC curve—receiver operating characteristic curve; AUC—area under the curve; PsA—psoriatic arthritis; GoA—gonarthrosis.

**Table 4 diagnostics-13-03250-t004:** Results of Spearman’s rank-order correlation analysis between disease activity indices and cathepsin G and cathepsin K in the serum and synovial fluid of the patients with PsA.

Parameters	VAS	DAPSA	PASDAI	mCPDAI	HAQ-DI
Cathepsin G, serum					
Correlation coef. rs	0.06	0.175 *	0.042	0.98	0.181 *
95% CI	(−0.07 to 0.21)	(0.19 to 0.31)	(−0.14 to 0.23)	(−0.05 to 0.22)	(0.19 to 0.42)
Significance (*p*)	0.366	0.021	0.389	0.176	0.024
Cathepsin K, serum					
Correlation coef. rs	0.04	0.189 *	0.067	0.185 *	0.04
95% CI	(−0.05 to 0.23)	(0.21 to 0.30)	(−0.1 to 0.19)	(0.24 to 0.31)	(−0.03 to 0.21)
Significance (*p*)	0.750	0.024	0.287	0.020	0.820
Cathepsin G, synovial fluid					
Correlation coef. rs	0.362 ***	0.191 **	0.067 *	0.378 ***	−0.011
95% CI	(0.24 to 0.53)	(0.21 to 0.30)	(0.1 to 0.19)	(0.03 to 0.19)	(−0.1 to 0.14)
Significance (*p*)	<0.001	0.001	0.022	<0.001	0.888
Cathepsin K, synovial fluid					
Correlation coef. rs	0.391 ***	0.182 *	0.384 ***	0.313 ***	0.354 ***
95% CI	(0.21 to 0.50)	(0.21 to 0.29)	(0.22 to 0.52)	(0.15 to 0.41)	(0.15 to 0.52)
Significance (*p*)	<0.001	0.02	<0.001	<0.001	<0.001

VAS—visual analog scale; DAPSA—disease activity in psoriatic arthritis; PASDAI—composite psoriatic arthritis disease activity indices; mCPDAI—composite psoriatic disease activity index; * significant at *p* < 0.05; ** significant at *p* < 0.01; *** significant at *p* < 0.001.

## Data Availability

Not applicable.

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
