# Peer review of "Serum and Synovial Levels of Cathepsin G and Cathepsin K in Patients with Psoriatic Arthritis and Their Correlation with Disease Activity Indices"

_diagnostics, 2023, doi:10.3390/diagnostics13203250_

Round 1

Reviewer 1 Report

Comments
1.    Minor English and grammar revisions are in order.
2.    Please check abbreviation management: abbreviate the first time in text and then use only the abbreviation.
3.    “Another insufficiently explored aspect is how crystals in PsA patients' synovial fluid affect the levels of Cathepsin G and Cathepsin K” – what crystals? Please state/define what crystals in the introduction in this phrase. Leaving “crystals” alone is rather contra intuitive since PsA is not a crystal-induced arthritis.
4.    Paragraphs from lines 67-76 and 77-82 on page 2, at the end of introduction, both state the study objectives. This is confusing and redundant: do you have 5 study objectives (3 in the first paragraph and 2 in the second one)? Probably not. Please merge in these paragraphs into a single paragraph in which to enunciate all your objectives.
5.    “the potential of serum Cathepsin G and Cathepsin K to distinguish PsA patients from gonarthrosis (GoA) patients” – a rheumatologist easily distinguishes PsA from GoA based on clinical interview, clinical examination, a few blood tests and some imaging information. You do not need cathepsis to make this distinction. So, there is no additional clinical value of measuring cathepsins for this. Therefore, why is it important to distinguish PsA from GoA using these complicated markers? This is an important question for your study objective. Do you aim to show that these cathepsins are more expressed in PsA compared to GoA? Please briefly update the text in this regard.
6.    Why did you choose knee osteoarthritis and not hand osteoarthritis? For practical reasons regarding the ease of joint aspiration? Please briefly state this in the introduction.
7.    “This was a retrospective case–control study, involving 156 patients with a PsA diagnosis according to the CASPAR criteria and 50 GoA patients.” – this is the starting phrase for the Methos section. The fact that you studied 156 PsAs and 50 GoAs is a result not a method. When you designed your study, you did not think to yourself: “hmm, we are going to enroll exactly 156 PsA patients!”. The fact that you end up having 156 PsA patients in your sample is a result of the study. So, please report the number of patients in the Results section.
8.    “This was a retrospective case–control study, involving 156 patients with a PsA diagnosis according to the CASPAR criteria and 50 GoA patients.” – we remind the authors that CASPAR is a set of CLASSIFCATION criteria for typical PsA cases. One does not diagnose PsA according to CASPAR. Your clinical diagnosis of PsA ALSO FULFILLS the CASPAR criteria. Please update text accordingly.
9.    Exclusion criterion 3 is “(3) absence of mental health comorbidities” – why is this relevant? What do anxiety or eating disorders have to do with PsA and cathepsins? They might have an impact, but please state it in the text, since it is not readily understandable.
10.    Regarding inclusion/exclusion criteria: did you enroll children as well? If not, please state that the age of 18 years was a inclusion/exclusion criterion.
11.    In the methods, the reader finds out that you also have a “healthy control group comprised individuals with no complaints and physical findings”. This is not mentioned in the study objectives. Please briefly include it.
12.    Materials and method is written as a hole big paragraph, which can be tiering to read. Please subdivide the “2. Materials and methods” into 4 sections: 2.1. patients and study groups, 2.2. cathepsin measurement protocol, 2.3. arthrocentesis and clinical evaluations, 2.4. statistics. Or in as many sections as you see fit, named whatever you want.
13.    “145 on the knee joint, 6 on the ankle joint, 3 on the lactic joint, and 2 on the shoulder joint” – I am sorry, I do not know what is the “lactic joint”. Please check if this is correct and update text accordingly.
14.    Who or how many authors/clinicians performed the joint counts necessary for the DAPSA? Please state this in the text since it impacts error of measurements.
15.    From the methods sections, the reader finds out that each PsA patient underwent the following procedures: questionnaires, clinical examination, venous puncture, arthrocentesis. Are these procedures performed in the same day for each patient? Please state this in the text since it impacts error of measurements.
16.    You have already published the information about crystals in a previous article on 156 PsA patients (Geneva-Popova M, Popova-Belova S, Popova V, Stoilov N. Assessment of Crystals in the Synovial Fluid of Psoriatic Arthritis Patients in Relation to Disease Activity. Diagnostics (Basel). 2022 May 18;12(5):1260. doi: 10.3390/diagnostics12051260. PMID: 35626414; PMCID: PMC9140193.). Your conclusions on this study are just about cathepsins. So, why not leave this article only about cathepsins? Unless you have a very important reason to leave crystals in. 
17.    In Table 1 you use the letters “f” and “t” to indicate footnotes. This is unusual. Please use standard symbols such as “*” and “#”.
18.    Table 1 differentiates the ages of women and men in the PsA subgroup but not in the GoA subgroup? Please be consistent for clarity: either report ages for men and women in the GoA group as well or not in both of them.
19.    In Table 1, the percentages of comorbidities in the GoA subgroup are mis-reported: “82.0026.0018.00”. Please correct this error.
20.    “The effect of the covariates was not significant: Cathepsin G (age p = 0.141; BMI p = 0.891).” – this phrase is rather short of words. Please add some words to explain what is it that you are trying to report.
21.    Figures 1 and 2. Please use different colors for the boxplots in PsA , GoA and healthy controls so that the image can by visually understood.
22.    Tables 2 and 3 report ROC analysis results. ROC is not mentioned in the statistics section. Please mention ROC in the statistics section.
23.    Table 4 reports Spearman (sic!) rank-order correlations which are not mentioned in the statistics section. Please do.
24.    The results section is missing its 3.2 sub-heading.
25.    Please add the limitations of your study at the end of the Discussion section.
26.    “Cathepsin” is a common noun. Please write it with a low case initial letter.
27.    References are disorganized. Please follow the journal’s indications about formatting the references.

please see previous recommendations.

Author Response

 Author's Notes to Reviewer â„–1

Author's Notes to Reviewer â„–1

  1. Minor English and grammar revisions are in order. - this has already been done
    2.    Please check abbreviation management: abbreviate the first time in text and then use only the abbreviation - this has already been done
    3.    “Another insufficiently explored aspect is how crystals in PsA patients' synovial fluid affect the levels of Cathepsin G and Cathepsin K” – what crystals? Please state/define what crystals are in the introduction of this phrase. Leaving “crystals” alone is rather intuitive since PsA is not a crystal-induced arthritis - It is true that psoriatic arthritis is not a crystalline arthropathy, but many researchers, including us, have published that 30% of patients with PsA have predominantly monosodium urate crystals. The presence of monosodium urate crystals in the synovial fluid of patients with PsA is associated with a more severe clinical course and a worse patient prognosis
    4.    Paragraphs from lines 67-76 and 77-82 on page 2, at the end of the introduction, both state the study objectives. This is confusing and redundant: do you have 5 study objectives (3 in the first paragraph and 2 in the second one)? Probably not. Please merge these paragraphs into a single paragraph in which to enunciate all your objectives- this has already been done
    5.    “the potential of serum Cathepsin G and Cathepsin K to distinguish PsA patients from gonarthrosis (GoA) patients” – a rheumatologist easily distinguishes PsA from GoA based on clinical interview, clinical examination, a few blood tests, and some imaging information. You do not need cathepsins to make this distinction. So, there is no additional clinical value in measuring cathepsins for this. Therefore, why is it important to distinguish PsA from GoA using these complicated markers? This is an important question for your study objective. Do you aim to show that these cathepsins are more expressed in PsA compared to GoA? Please briefly update the text in this regard. It is true that there are different methods for distinguishing PsA from gonarthrosis. The aim of our report is to prove that the studied cathepsins are in greater quantity in patients with PsA, respectively this condition is more destructive and debilitating.

  2. Why did you choose knee osteoarthritis and not hand osteoarthritis? For practical reasons regarding the ease of joint aspiration? Please briefly state this in the introduction. – We chose to study a comparison of PsA and gonarthrosis because the knee joint is easier to study, with more synovial fluid and easy access for arthrocentesis

  3. “This was a retrospective case–control study, involving 156 patients with a PsA diagnosis according to the CASPAR criteria and 50 GoA patients.” – this is the starting phrase for the Methos section. The fact that you studied 156 PsAs and 50 GoAs is a result, not a method. When you designed your study, you did not think to yourself: “hmm, we are going to enroll exactly 156 PsA patients!”. The fact that you end up having 156 PsA patients in your sample is a result of the study. So, please report the number of patients in the Results section. - - this has already been done

    8.    “This was a retrospective case–control study, involving 156 patients with a PsA diagnosis according to the CASPAR criteria and 50 GoA patients.” – we remind the authors that CASPAR is a set of CLASSIFCATION criteria for typical PsA cases. One does not diagnose PsA according to CASPAR. Your clinical diagnosis of PsA ALSO FULFILLS the CASPAR criteria. Please update the text accordingly.- - this has already been done
    9.    Exclusion criterion 3 is “(3) absence of mental health comorbidities” – why is this relevant? What do anxiety or eating disorders have to do with PsA and cathepsins? They might have an impact, but please state it in the text, since it is not readily understandable. - This criterion is included so that we can be sure that patients understand the meaning of their participation in the study. It is not ethical for patients with mental disorders to participate in research.
  4. Regarding inclusion/exclusion criteria: did you enroll children as well? If not, please state that the age of 18 years was an inclusion/exclusion criterion. this has already been done

  5. In the methods, the reader finds out that you also have a “healthy control group comprised individuals with no complaints and physical findings”. This is not mentioned in the study objectives. Please briefly include it. - it is mentioned

  6. Materials and method is written as a whole big paragraph, which can be tiring to read. Please subdivide the “2. Materials and methods” into 4 sections: 2.1. patients and study groups, 2.2. cathepsin measurement protocol, 2.3. arthrocentesis and clinical evaluations, 2.4. statistics. Or in as many sections as you see fit, named whatever you want.- this has already been done

  7. “145 on the knee joint, 6 on the ankle joint, 3 on the lactic joint, and 2 on the shoulder joint” – I am sorry, I do not know what is the “lactic joint”. Please check if this is correct and update the text accordingly. - the text is wrong, only knee joints are analyzed in the corrected text

  8. Who or how many authors/clinicians performed the joint counts necessary for the DAPSA? Please state this in the text since it impacts the error of measurements. The examination was performed by two independent rheumatologists

  9. From the methods sections, the reader finds out that each PsA patient underwent the following procedures: questionnaires, clinical examination, venous puncture, and arthrocentesis. Are these procedures performed on the same day for each patient? Please state this in the text since it impacts the error of measurements. Yes, all tests are performed on the same day

  10. You have already published the information about crystals in a previous article on 156 PsA patients (Geneva-Popova M, Popova-Belova S, Popova V, Stoilov N. Assessment of Crystals in the Synovial Fluid of Psoriatic Arthritis Patients in Relation to Disease Activity. Diagnostics (Basel). 2022 May 18;12(5):1260. doi: 10.3390/diagnostics12051260. PMID: 35626414; PMCID: PMC9140193.). Your conclusions on this study are just about cathepsins. So, why not leave this article only about cathepsins? Unless you have a very important reason to leave crystals in. - the authors' decision was to divide the patients into two groups - those with crystals and those without crystals

  11. In Table 1 you use the letters “f” and “t” to indicate footnotes. This is unusual. Please use standard symbols such as “*” and “#”. This is the authors' decision

  12. Table 1 differentiates the ages of women and men in the PsA subgroup but not in the GoA subgroup. Please be consistent for clarity: either report ages for men and women in the GoA group as well or not in both of them. - this has already been done

  13. In Table 1, the percentages of comorbidities in the GoA subgroup are misreported: “82.0026.0018.00”. Please correct this error. - this has already been done

  14. “The effect of the covariates was not significant: Cathepsin G (age p = 0.141; BMI p = 0.891).” – this phrase is rather short of words. Please add some words to explain what is it that you are trying to report. - According to the obtained results, age and BMI do not influence the level of cathepsin G.

  15. Figures 1 and 2. Please use different colors for the boxplots in PsA , GoA, and healthy controls so that the image can be visually understood. - these are the capabilities of the computer program
    22.    Tables 2 and 3 report ROC analysis results. ROC is not mentioned in the statistics section. Please mention ROC in the statistics section. - this has already been done

  16. Table 4 reports Spearman (sic!) rank-order correlations which are not mentioned in the statistics section. Please do. this has already been done

  17. The results section is missing its 3.2 sub-headings. - this has already been done

  18. Please add the limitations of your study at the end of the Discussion section.- this has already been done

  19. “Cathepsin” is a common noun. Please write it with a low case initial letter.- this has already been done

  20. References are disorganized. Please follow the journal’s indications about formatting the references.- this has already been done

Reviewer 2 Report

Article ID: DIAGNOSTICS 2600145

Title: Serum and synovial cathepsin G and cathepsin K levels in patients with psoriatic arthritis their relation to disease activity indices

Authors:   Mariela Gencheva Geneva-Popova,Stanislava Dimitrova Popova-Belova ,Krasimir Iliev Kraev, Velichka Zaharieva Popova.

General comment

In this manuscript authors carried out a retrospective case–control study, involving 156 patients, in which they examined the relationship between the serum and synovial cathepsin G and cathepsin K in patients with psoriatic arthritis compared with gonarthrosis (GoA) patients and control health subjects. Results of serum and synovial levels of cathepsin G and cathepsin K in patients with PsA compared to  control group of GoA patients showed that the levels of cathepsins were higher in the serum and synovial fluid of patients with PsA. In light of these data authors suggests that this could be associated with more severe damage of cartilage in PsA patients  with respect to GoA patients and this measurement could be used as a diagnostic biomarker to distinguish the two diseases.

The focused topic is of interest and the experiments appear to have been carried out in a sufficient way. However, there are some unclear points that need to be explored further. Therefore I have some questions and suggestions that should be addressed.

Specif points

            1) Since these are inflammatory diseases, it would have been useful

           to also consider some inflammatory cytokines such as TNF-alpha, IL-6

           and IL-1beta.

2) The study, as presented, seems poor and could be enriched with a comparison between inflammatory markers and the two cathepsins.

 3) It is advisable to state in the ABS that this is a retrospective study as it is not clear.

              4) The "Discussion" section should be expanded and improved.

5) It would be useful to include in the text a scheme showing the biochemical pathways of activity of the two cathepsins involved in cartilage damage in the two types of arthritis considered.

 6) Many sentences should be rewritten/checked by an English speaker.

No comment

Author Response

Reviewer #2

            Thank You very much for the critical notes. Some texts have been corrected, others cannot be corrected for objective reasons.

            Answers are marked in red.

Specific points

1) Since these are inflammatory diseases, it would have been useful to also consider some inflammatory cytokines such as TNF-alpha, IL-6, and IL-1beta - Pro-inflammatory cytokines were not examined in this study, so they cannot be discussed. A future study will take this into account.

2) The study, as presented, seems poor and could be enriched with a comparison between inflammatory markers and the two cathepsins. - The study's findings suggest that elevated levels of these cathepsins are associated with higher disease activity, as indicated by disease activity indices. This information underscores the potential clinical relevance of monitoring Cathepsin G and Cathepsin K levels as biomarkers for disease activity in PsA. Early detection of higher cathepsin levels may indicate the need for more aggressive and timely treatment. This could be valuable information for rheumatologists and healthcare providers involved in managing patients with PsA, as it may help guide treatment decisions and improve patient outcomes.

 3) It is advisable to state in the ABS that this is a retrospective study as it is not clear.- this has already been done

4) The "Discussion" section should be expanded and improved.- this has already been done

5) It would be useful to include in the text a scheme showing the biochemical pathways of activity of the two cathepsins involved in cartilage damage in the two types of arthritis considered.- this has already been done

 6) Many sentences should be rewritten/checked by an English speaker.- this has already been done

Reviewer 3 Report

Reviewed manuscript show interesting and well presented data. The introduction is good and statistic is impeccable. Discussion is compact and to the point. However, it lacks a clear emphasis on what is there new –it should be added both in the summary and in the abstract, because the presented discussion refers to numerous literature on similar topics. Of the minor corrections: in the abstract, there is no explanation of the acronym GoA, and line 119 in methodology - the division into paragraphs should be clearly marked.

Author Response

Reviewer #3

            Thank You very much for the critical notes.

            Answers are marked in red.

1. In the summary there is no explanation of the abbreviation GoA - it has been done

2. Line 119 in the methodology - the division into paragraphs must be clearly indicated - it has been done

Round 2

Reviewer 1 Report

The authors have addressed the important issues raised in a rather satisfying manner.

Reviewer 2 Report

Although some of the Referee's requests included the addition and comparison of further data through new experiments, the authors preferred to respond that they will verify the request in a future experiment. The other requests were granted. The manuscript can be accepted in this form.